# Cabotegravir Exposure of Zebrafish (*Danio rerio*) Embryos Impacts on Neurodevelopment and Behavior

**DOI:** 10.3390/ijms24031994

**Published:** 2023-01-19

**Authors:** Daniela Zizioli, Isabella Zanella, Luca Mignani, Melania Degli Antoni, Francesco Castelli, Eugenia Quiros-Roldan

**Affiliations:** 1Department of Molecular and Translational Medicine, University of Brescia, 25123 Brescia, Italy; 2Clinical Chemistry Laboratory, Cytogenetics and Molecular Genetics Section, Diagnostic Department, ASST Spedali Civili di Brescia, 25123 Brescia, Italy; 3Division of Infectious and Tropical Diseases, ASST Spedali Civili di Brescia, 25123 Brescia, Italy; 4Department of Clinical and Experimental Sciences, University of Brescia, 25123 Brescia, Italy

**Keywords:** cabotegravir, dolutegravir, integrase inhibitor, INSTI, pregnancy, developmental toxicity, zebrafish embryo

## Abstract

As most new medications, Cabotegravir (CAB) was recently approved as an antiretroviral treatment of HIV infection without in-depth safety information on in utero exposure. Although no developmental toxicity in rats and rabbits was reported, recent studies demonstrated that CAB decreases pluripotency of human embryonic stem cells. CAB exposure effects during development were assessed in zebrafish embryos by the Fish Embryo Toxicity test after exposure at subtherapeutic concentrations up to 25× the human C_max_. Larvae behavior was assessed by the light–dark locomotion test. The expression of factors involved in neurogenesis was evaluated by whole-mount in situ hybridization. CAB did not cause gross morphological defects at low doses, although pericardial edema, uninflated swim bladder, decreased heartbeats, growth delay, and decreased hatching rate were observed at the highest concentrations. Decreased locomotion was observed even at the subtherapeutic dose, suggesting alterations of nervous system integrity. This hypothesis was supported by the observation of decreased expression of crucial factors involved in early neuronal differentiation in diencephalic and telencephalic dopaminergic areas, midbrain/hindbrain boundary, and craniofacial ganglia. These findings support CAB effects on neurogenesis in zebrafish embryos and suggest long-term follow-up of exposed infants to provide data on drug safety during pregnancy.

## 1. Introduction

The HIV integrase strand transfer inhibitor (INSTI) class is globally one of the preferred options in HIV treatment guidelines. Within this class, Cabotegravir (CAB) is the newest drug approved for clinical use in HIV-infected patients. It belongs to the carbamoyl pyridone class, and it is an analog of dolutegravir (DTG), another INSTI widely used around the world. CAB shows optimal potency at low concentrations, with a significant −2.2 to −2.5 log10 HIV-1 RNA reduction after a 10-day monotherapy trial with 5–30 mg daily oral dosing, and has a protein binding-adjusted 90% inhibitory concentration (IC90) of 166 ng/mL [1]. Unlike DTG, CAB has the unique feature of a long half-life and can be formulated as nanoparticles coated with Polyethylene Glycol, with the possibility of being administered at prolonged intervals. Its long-acting formulation is dosed monthly or every 2 months. CAB also differs from DTG in that it inhibits the Organic Anion Transporters 1 (OAT1) and 3 (OAT3), whereas DTG inhibits OAT2 [2,3].

Regarding DTG, conflicting results about its safety have been published. Initially, the Tsepamo study suggested an increase in neural tube defects in the fetus after DTG exposure during pregnancy in humans, but subsequent in vitro and in vivo studies showed conflicting results regarding the role of DTG in inducing neural tube defects [4,5,6,7,8,9,10,11]. However, a study including children who were HIV-exposed but uninfected and exposed in utero to antiretroviral medications suggested an association between DTG exposure and an increased risk of having neurologic abnormalities in infancy and childhood [12].

As most new medications, CAB was approved for use in HIV-infected adults without long-term real-life safety data or safety information on exposure during pregnancy or considering infants exposed in utero. Notwithstanding, the European Medicines Agency (EMEA) assessment report on CAB has described results from non-clinical studies about genotoxicity, male and female fertility, and embryo-fetal developmental toxicity using two mammalian species as models. Embryo-fetal toxicity studies were conducted in rats and rabbits at doses up to 2000 mg/kg/day, and there was neither a treatment-related increase in post-implantation loss nor treatment-related increase in fetal malformations at any dose level, although in rats but not in rabbits a decrease in fetal weight at the high dose level was observed [13]. Nonetheless, given the high similarity in their chemical structure between CAB and DTG, and considering that pregnant women are usually underrepresented or excluded in drug clinical trials, there is an urgent need to explore any possible associations between CAB exposure and neurodevelopmental toxicity in animal models.

Zebrafish (*Danio rerio*) is a small vertebrate used as a model organism for studies on developmental and toxicological effects of chemicals and drugs [14]. The development of embryos is rapid, with organogenesis completed within 3–4 days post-fertilization (dpf). In addition, zebrafish share a high degree of genetic similarity, functional homology, physiological, and developmental processes with humans [14,15]. After hatching, occurring at 48–72 h post fertilization (hpf), larvae can be used to assess locomotor activity due to their spontaneous swimming by 4–5 dpf, and behavioral tests can be used to monitor their responses to neuroactive compounds [16].

Here, we assessed the effect of CAB exposure during embryonic development using the zebrafish embryo as a model.

## 2. Results

### 2.1. Mortality Assessment

We first assessed whether CAB may exert toxic effects in zebrafish embryos by evaluating survival after direct exposure by the immersion method. According to the OECD TG 236, mortality was around 10% for negative (solvent) control and higher than 30% in the DCA positive control at all time endpoints (Appendix A). For doses of CAB lower or equal to the human C_max_, mean survival rates of treated embryos were not significantly different when compared with those observed for negative controls at each time endpoint (Figure 1), suggesting no or very low toxicity of CAB in the human therapeutic range. However, embryos showed significantly lower survival rates at higher doses. CAB microinjection experiments confirmed the results obtained using water-treatment protocol for survival rate at different developmental stages (Appendix A).

### 2.2. Morphological Assessments and Hatching Rate

To assess whether CAB may induce developmental malformations, a thorough evaluation of morphological parameters was performed on embryos treated by the immersion method. No evident malformations were observed at all tested doses at 96 hpf, except for a mild pericardial edema (25.5% of embryos) and either undeveloped or uninflated swim bladder (9.5% of embryos) at the highest dose (500 μM), with 6.7% of embryos showing both defects (Figure 2).

We also noted a slight body length shortening, starting at the 100 µM dose. Similar defects were observed at 120 hpf. At this experimental endpoint (120 hpf), we also measured body length and weight, finding significant body length shortening of embryos treated with CAB 100 and 500 μM (67.0% and 72.0% of embryos, respectively) and a decrease of the average embryo weight at the same doses (69.0% and 72.0% of embryos, respectively) (Figure 3A,B). At the same time endpoint and doses, we also measured slightly decreased heartbeats (about 70.0% and 76.0% of embryos, respectively, with CAB 100 and 500 μM) (Figure 3C).

Gross morphology was also carefully analyzed in microinjected embryos, and similar defects were observed at 96 hpf, however already starting from 100 μM CAB exposure (2.7 and 13.0% of embryos with pericardial edema, 9.2 and 15.2% with undeveloped or uninflated swim bladder, 4.2 and 5.0% with both defects, at 100 and 500 μM, respectively) (Appendix A), and again with no further morphological worsening at 120 hpf.

No significant effects on hatching were observed up to the 50 µM CAB dose, whereas slightly but significantly decreased hatching rates were observed at the highest doses with both types of drug exposure (Figure 4 and Appendix A).

### 2.3. Neurotoxicity Evaluation

Since hatching needs movement of embryos, we assessed swimming behavior of in-water exposed embryos at 144 hpf by a light–dark locomotion test. The analysis of the total (light + dark) distance swam by the larvae revealed that locomotor activity was severely reduced by CAB exposure, already from the lowest doses (Figure 5A). Exposed larvae similarly decreased their movements in both dark and light conditions and showed reduced movement speed (Figure 5B), suggesting potential effects of the drug on neurodevelopment although no evident alteration of brain gross morphology.

We investigated this hypothesis by performing WISH experiments with two different neuro-specific riboprobes, *neurod1* and *neurog1*. CAB exposure resulted in decreased *neurod1* expression at 96 hpf in a large percentage of treated embryos (89.0 and 94.0%, respectively, for 10 and 20 μM CAB exposed embryos), as evidenced by several areas less intensively decorated by the specific riboprobe in comparison with control embryos (Figure 6A). Specifically, we observed reduced *neurod1* expression in regions corresponding to the diencephalon, midbrain/hindbrain boundary, and craniofacial ganglia, with *neurod1* mRNA almost completely absent at the dose of 20 μM in the diencephalic area, suggesting altered neuronal differentiation in these areas. To deepen these results, we further investigated the expression at 16 hpf of *neurog1*. CAB exposure resulted in decreased *neurog1* expression in the diencephalic and telencephalic dopaminergic areas and in cranial ganglia, with a high percentage of treated embryos revealing a downregulated *neurog1* expression (87.0 and 92.0%, respectively, for 10 and 20 μM exposed embryos) (Figure 6B).

## 3. Discussion

We studied the safety of CAB in the zebrafish embryo model by evaluating the phenotype during development until 144 hpf after drug exposure at doses ranging from subtherapeutic concentrations (10 µM) up to 500 µM (25X C_max_). CAB does not seem to affect the survival rate of zebrafish larvae, nor does it cause gross morphological defects up to concentrations of 50 µM, although at above concentrations, pericardial edema, undeveloped or uninflated swim bladder, decreased heartbeats, growth delay, and decreased hatching rate were observed. We also observed a decreased locomotion in zebrafish larvae exposed to CAB, even at low doses (10 and 20 µM, subtherapeutic and therapeutic doses, respectively), suggesting a possible alteration of the nervous system integrity that was further confirmed by demonstrating defects on neuronal differentiation.

Previous data from studies on embryo-fetal CAB safety performed in rats and rabbits did not disclose any alterations, except for a decrease in the offspring weight [13]. More recently, a first analysis evaluating pregnancy outcomes in 25 women exposed to CAB at the time of conception found a rate of spontaneous abortions of 24%; however, this may be explained considering that CAB-treated women were more frequently tested for pregnancy in order to discontinue the treatment and switch to an alternative antiviral therapy and, as a consequence, very early pregnancies (then with a higher risk of abortion) were identified [17]. Typically, pregnant women are excluded from enrollments in drug clinical trials and women of childbearing potential are required to use effective method of contraception. Therefore, animal or in vitro models may be useful tools to study drug embryotoxicity and choose the clinically relevant dosing in these models. A recent pharmacokinetic study in a mouse pregnancy model found that plasma human C_max_ concentrations for CAB were achieved in pregnant mice at a 10X dose of CAB [18]. This study also showed a low placental transfer for CAB, with drug concentrations in the amniotic fluid below the reported therapeutic concentrations. No data were however reported in this study regarding pregnancy outcomes. Further studies demonstrated that, even at subtherapeutic concentrations, CAB, as well as other second-generation INSTIs, like bictegravir and DTG, but not first-generation INSTIs, like raltegravir, can induce cytotoxicity, loss of pluripotency, and dysregulation of genes involved in early differentiation in human embryonic stem cell lines, although no effects on litter size or fetal weight was observed in a pregnancy mouse model after CAB exposure at the human C_max_ dose [19].

Long-acting CAB is nowadays not recommended during pregnancy, due to there not being enough data on safety in women who become pregnant and in their foetuses and babies. Therefore, there is an urgent need for data about the safety of antiretroviral drugs like CAB during pregnancy [20].

In our embryo model, although no evident gross morphology defects were found in the brain area, significant alterations in the expression pattern of *neurog1* and *neurod1* genes were detected at the level of the midbrain-hindbrain boundary, ventral diencephalic region, telencephalon, and craniofacial ganglia, at both therapeutic (20 µM) and subtherapeutic doses (10 µM) and even from the very early stages of development. *Neurog1* is one of the transcription factors expressed at the very onset of neurogenesis in the zebrafish neural plate during late gastrulation, is required for the development of dopaminergic neurons of the forebrain, is transiently detectable in telencephalic and ventral diencephalic neurons and in cranial ganglia, and is an upstream regulator of *neurod1*, that is in turn detectable at later stages of embryo development [21,22,23]. *Neurod1* gene is a further key differentiation factor during neurogenesis, mainly expressed in differentiated neurons of the midbrain and hindbrain, midbrain-hindbrain boundary, and cranio-facial ganglia [21]. The evident downregulation of both transcription factors after CAB exposure in zebrafish embryos suggests early effects of the drug on neuronal differentiation and brain development that could reflect in the observed delayed embryo hatching, since hatching needs embryo movement, and, after the development of the swim bladder at 96–120 hpf, in the documented deficit in motorial activities of the exposed larvae.

A recently published study could help to suggest a possible explanation for our results. Bade and colleagues [24] have recently described DTG as a broad-spectrum matrix metalloproteinases (MMPs) inhibitor by binding to Zn^++^ at the catalytic domain of these enzymes. Interestingly, MMPs inhibition following DTG exposure during gestation seemed to induce neuroinflammation and impair pre- and post-natal neurodevelopment in mice. Using computational molecular docking models, the same authors demonstrated that other INSTIs (raltegravir, bictegravir, and also CAB) possess chemical structures prone to bind Zn^++^ at the catalytic domain of MMPs-, suggesting that those drugs may have chemical abilities for broad-spectrum MMPs inhibition, like DTG. This might be of even greater significance for CAB, considering its high chemical similarity with DTG, although this hypothesis is merely speculative and further studies are needed to demonstrate the hypothesized effects. The MMPs, a family of structurally related Zn-dependent endoproteinases, play a key role in the processes of neurodevelopment, regulating multiple activities like angiogenesis, neurovasculature remodeling, establishment, and integrity of the blood–brain barrier (BBB), neurogenesis, neuronal migration, myelination, axonal guidance, synaptogenesis, synaptic plasticity, and tissue remodeling [25,26]. Alternatively, CAB may act as a Zn-chelator. It is indeed well documented that gestational Zn deficiency in humans results in deleterious effects on brain development and functions, leading to alterations in neonate behavior, with impaired cognitive and motor performance [27].

Clearly, our study presents some limitations. Despite the great similarities between zebrafish and mammals in terms of anatomy, physiology, genetics, biochemistry, and development, regardless of the evolutionary distance, questions remain about how relevant toxicity studies in zebrafish are to humans [14]. Secondly, we did not determine the exact concentration of CAB taken up by embryos in immersion experiments. However, toxicity tests in zebrafish embryos are usually performed through this method, which is based on diffusion through the skin, since the mouth opens from 3 dpf and gills are not functional till 14 dpf [14]. We are aware that in-water dosing may not exactly match human plasma levels, although overall chemical absorption by zebrafish embryos is usually higher for poorly water-soluble and small molecules like CAB. Furthermore, even considering low absorption through embryo skin, we observed locomotor dysfunctions and low expression of factors that are crucial for neurogenesis even at the 10 µM concentration, suggesting that CAB may act on this pathway even at subtherapeutic concentrations. This can be of importance, considering that placental transfer of CAB seems to be low in an ex-vivo human cotyledon perfusion model [28]. Moreover, to overcome the question of a sufficient uptake of the drug, we also microinjected the molecule and observed similar morphological effects, which suggests sufficient uptake from water.

## 4. Materials and Methods

### 4.1. Ethics Statement

All animal experiments were conducted in accordance with the Italian and European regulations on animal care and the standard rules defined by the Local Committee for Animal Health (Organismo per il Benessere Animale) and authorized by the Italian Ministry of Health (Authorization Number 287/2018).

### 4.2. Zebrafish Maintenance and Collection of Eggs

Wild type adult animals (AB strain) were bred in a recirculating aquaculture system (Techniplast ZebTEC, Buguggiate, VA, Italy), maintained at 28.5 °C in a 14 h light and 10 h dark daily cycle, at the Zebrafish Facility of the University of Brescia, essentially as previously described by Zizioli and colleagues [29]. Adult male and female fish were mated in the mating box overnight. Freshly spawned eggs were collected the next morning, washed with fresh zebrafish water (0.1 g/L Instant Ocean Sea Salts, 0.1 g/L Sodium Bicarbonate, 0.19 g/L Calcium Sulfate), and maintained at 28 °C in Petri dishes containing fresh fish water in 14–10 h light-dark cycle until exposure to CAB. Embryo staging was done as described by Kimmel and colleagues [30]. All experiments were conducted exposing embryos to tested solutions at the gastrula stage (4 hpf) up to 48, 72, 96, 120, or 144 hpf, as described in each experiment.

### 4.3. Drug Exposure of Embryos

A stock solution of 1 mM CAB was prepared by dissolving the drug (Merck Life Science, catalog # AMBH2D6FBF49) in water plus dimethyl sulfoxide (DMSO, 10% final concentration) (Merck Life Science, Darmstadt, Germany). Exposure solutions were freshly prepared by serial dilutions of the stock in fish water with 0.1% as final concentration of solvent (DMSO). In experiments conducted by the immersion method, alive embryos were dechorionated to maximize drug uptake and transferred to glass Petri dishes used as exposure containers. Dechorionated embryos were exposed to CAB doses in the 10–500 µM range (10–20–50–100–500 µM). Only for hatching rate, non-dechorionated eggs were used. The 10–500 µM dose range was selected starting approximately with the in vitro C_max_ (9.74 µM), comprising the human therapeutic dosage (C_max_ 8 µg/mL, corresponding to about 20 µM) and with higher doses up to a large multiple (25×) of the human C_max_. As negative control, embryos were exposed to 0.1% DMSO in fish water (expected mortality rate < 10%). As positive control for survival rate experiments, we used 3,4-dichloroaniline (DCA) (Sigma-Aldrich, Saint Louis, MO, USA) dissolved in fish water at the concentration of 3.74 mg/L (expected mortality rate > 85–90%) [31]. To ensure that the drug was sufficiently taken up by embryos, further experiments were conducted by direct drug microinjection. Embryos were injected at 1–2 cell stage with the same concentrations of CAB used in immersion experiments, essentially as previously described [32]. Briefly, the total volume of injection was 5 nL/embryo, and 50 embryos were injected for each drug dose or controls. CAB was co-injected with 0.05% phenol red as tracer and control for the microinjection process. Embryos injected with 0.05% phenol red in sterile water and non-injected embryos were used as controls. After microinjection, embryos were collected in Petri dish and maintained in fish water at 28 °C until further evaluations.

### 4.4. Evaluation of Mortality, Gross Morphology, and Hatching Rate

Drug-induced mortality and embryotoxicity were evaluated by the Fish Embryo Toxicity test (FET), essentially as previously described [29,31]. For experiments performed by both direct immersion and microinjection, mortality was evaluated at 48, 72, 96, and up to 120 hpf. A dose-response survival graph was plotted for each dose and time endpoint. For both types of drug exposure, morphology was carefully evaluated up to 120 hpf by daily visual inspection of anesthetized embryos (0.4% Tricaine) (Sigma-Aldrich) from head to tail, under microscope direct visualization with Zeiss Axiozoom V13 microscope (Zeiss, Oberkochen, Germany), equipped with PlanNeoFluar Z 1×/0.25 FWD 56 mm lens and ZEN3.5 (Blu version) software (Zeiss) (magnification 20×). As morphological endpoints, we considered: body length, body weight, eye dimension, head and tail morphology, anterior-posterior (AP) axis, absence/presence of pericardial edema, absence/presence of the inflated swim bladder, yolk sac morphology, and dimension. The time endpoint of 96 hpf for image reporting was selected on the basis that all organs are completely developed at that stage. For immersion experiments, body length measurements were performed at 120 hpf on digital images using the Image J Fiji version 2.9.0; weight was measured at the same stage as the average weight of 30 embryos per experiment; heart beats at the same stage were manually counted for 1 min under Leica stereomicroscope in quiet conditions and at optimal temperature. Hatching rate was recorded at 72 and 96 hpf. For all the above experiments, we used 30 (for immersion experiments) or 50 embryos (for injection experiments) for each experiment and each treatment or control. All experiments were repeated three times.

### 4.5. Behavior Assessment by the Light-Dark Locomotion Test

Embryos at 4 hpf were exposed until 120 hpf to CAB or fish water with 0.1% DMSO as control, by the immersion method. Experiments were repeated three times. For each treatment and each experiment, 24 larvae were then collected in a 96-square well plate with one larva per well in a volume of 200 µL (treatment or fish water with 0.1% DMSO) and grown until 144 hpf at 28 °C with a light–dark cycle. The plate was then placed in the observation chamber of the Danio Vision (Noldus) system holder in an isolated noise free room. Larvae were acclimatized for 30 min before video recording. The system was set up to track movements (moved distance in 2 min time bins) for 2 h by 6 cycles of alternating light and dark 10 min periods. Data were analyzed using the Noldus Ethovision software (Noldus, Wageningen, Nederland). Movements were reported as total distance (cm) travelled by the larvae, calculated during both light and dark stimuli. Speed (mm/s) was also calculated by the same software.

### 4.6. Whole-Mount In Situ Hybridization (WISH)

DIG-labelled antisense RNA probes for neurogenin1 (neurog1) and neuronal differentiation 1 (neurod1) genes were prepared by in vitro transcribed linearized cDNA clones with T7 and SP6 polymerase using Digoxigenin Labeling Mix (Roche, Basel, Switzerland), essentially as previously described [33]. Whole-mount in situ hybridization (WISH) was performed according to standard methods [33]. Briefly, treated embryos (16 hpf or 96 hpf, respectively, for neurog1 and neurod1 expression evaluations) were fixed overnight in 4% (*v*/*v*) paraformaldehyde (Sigma-Aldrich) at 4 °C, dehydrated through an ascending methanol (Sigma-Aldrich) series, and stored at −20 °C. After permeabilization with 10 µg/mL proteinase K (Roche), embryos were hybridized overnight at 68 °C with DIG-labelled antisense RNA probes (400 ng). After several washes at high stringent temperature with SSC 2X/PBS and SSC 0·2X/PBS, embryos were incubated with an anti-digossigenin (DIG) antibody conjugated with alkaline phosphatase (Roche) overnight at 4 °C. Staining was performed with NBT/BCIP (Roche) alkaline phosphatase substrates, according to the manufacturer’s instructions. Embryos were mounted in agarose-coated dishes and WISH images were taken with a Leica MZ16F stereomicroscope equipped with DFC 480 digital camera and LAS Leica Imaging software (Leica Microsystems, Microsystems, Wetzlar, Germany) at 20× magnification. Experiments were repeated twice with 25 embryos for each treatment and each experiment.

### 4.7. Statistical Analysis

Data are presented as mean ± SD. All graphs were plotted using GraphPad Prism 8. Significance was analyzed by Student’s *t* test. *p*-values less than 0.05 were considered as significant (* *p* < 0.05; ** *p* < 0.005).

## 5. Conclusions

In conclusion, although the deep mechanisms involved in the zebrafish neurodevelopmental impairment following CAB exposure need to be thoroughly explored, our study provides data about safety of CAB on zebrafish embryo development, may serve to advance translational research in evaluating the effects of exposure to CAB on embryo neurogenesis, and highlights the need to study the long-term neurological effects in children exposed to CAB during fetal life [12].

## Figures and Tables

**Figure 1 ijms-24-01994-f001:**
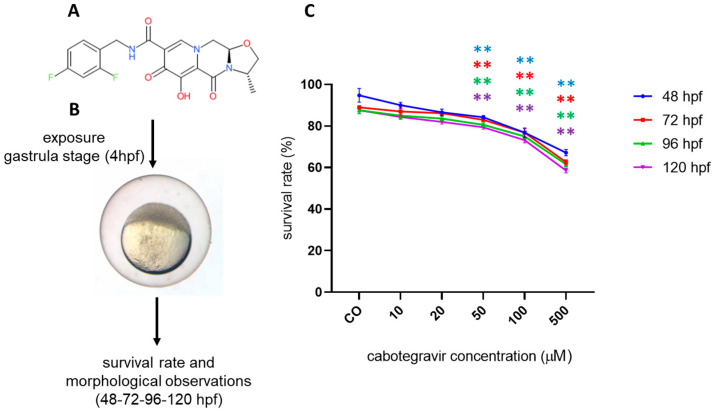
Survival rate of zebrafish embryos at different developmental stages (hours post fertilization, hpf) after exposure to cabotegravir (CAB) by the immersion method. (**A**) CAB chemical structure. (**B**) Overall experimental design. (**C**) Dechorionated zebrafish embryos were exposed at gastrula stage (4 hpf) to drug solvent only (fish water plus 0.1% dimethyl sulfoxide, DMSO) [CO] or CAB, dissolved in fish water containing 0.1% DMSO. *X*-axis shows drug doses used for exposure of embryos; *Y*-axis shows the corresponding survival percentages at the different developmental stages (48 hpf, blue line; 72 hpf, red line; 96 hpf, green line; 120 hpf, violet line). Results are expressed as mean ± SD of three independent experiments, with 30 embryos for each experiment and each treatment. (** *p* < 0.005 vs. control group).

**Figure 2 ijms-24-01994-f002:**
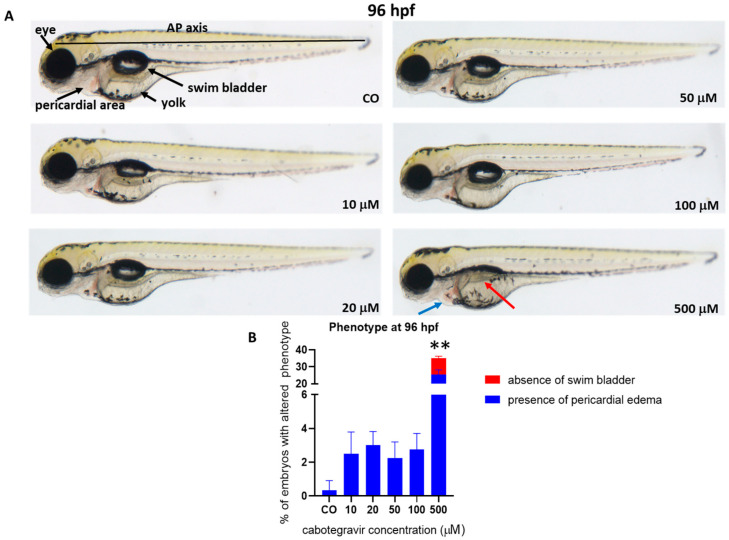
Representative pictures at 96 h post fertilization (hpf) of the gross morphological effects on zebrafish embryos of cabotegravir (CAB) exposure by the immersion method. (**A**) Dechorionated embryos were exposed by the immersion method to different CAB doses from gastrula stage (4 hpf) up to 96 hpf as time endpoint for morphology observations. CAB was dissolved in fish water containing 0.1% dimethyl sulfoxide, DMSO. Control embryos [CO] were treated with fish water and 0.1% solvent (DMSO). All treatments including controls were conducted three times, with 30 embryos for each experiment and each treatment. Embryos in the figures are representative of all experiments. All pictures are lateral views with dorsal to the top and anterior to the left. The red arrow indicates the absence of an inflated swim bladder, and the blue arrow indicates the presence of pericardial edema. (**B**) Percentages of embryos with defects. Results are expressed as mean ± SD of three independent experiments, with 30 embryos for each experiment and each treatment. (** *p* < 0.005 vs. control group).

**Figure 3 ijms-24-01994-f003:**
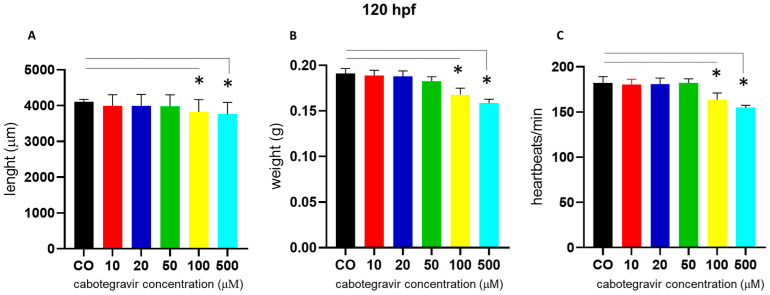
Effects of cabotegravir (CAB) exposure of zebrafish embryos (immersion method) on length, weight, and heartbeat at 120 h post fertilization (hpf). Dechorionated embryos were exposed by the immersion method to different drug doses at gastrula stage (4 hpf) up to 120 hpf. CAB was dissolved in fish water containing 0.1% dimethyl sulfoxide (DMSO). Control embryos [CO] were treated with fish water and 0.1% DMSO. *X*-axis shows drug doses used for exposure of embryos; *Y*-axis shows: (**A**) length (µm), (**B**) weight (g), and (**C**) heartbeats/min. Results are expressed as mean ± SD of three independent experiments, with 30 embryos for each experiment and each treatment. (* *p* < 0.05 vs. control group).

**Figure 4 ijms-24-01994-f004:**
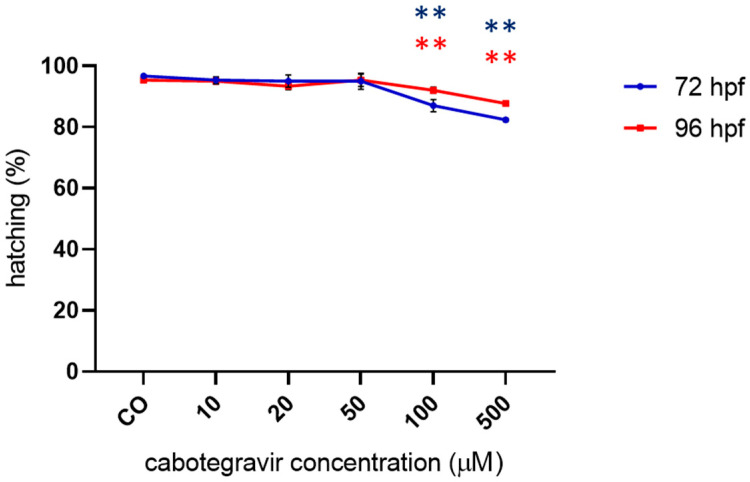
Percentages of hatched embryos at 72 and 96 h post fertilization (hpf) after exposure to cabotegravir (CAB) by the immersion method. Non-dechorionated embryos were exposed by the immersion method at the gastrula stage (4 hpf) and up to 72 and 96 hpf to different drug doses. CAB was dissolved in fish water containing 0.1% dimethyl sulfoxide (DMSO). Control embryos [CO] were treated with fish water and 0.1% DMSO. X-axis shows drug doses used for exposure of embryos; Y-axis shows the corresponding percentages of hatched embryos at 72 (blue line) and 96 (red line) hpf. Results are expressed as mean ± SD of three independent experiments, with 30 embryos for each experiment and each treatment. (** *p* < 0.005 vs. control group).

**Figure 5 ijms-24-01994-f005:**
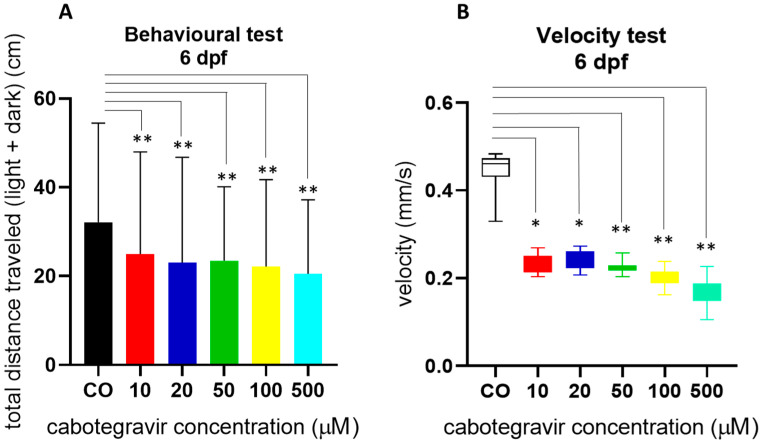
Swimming behavior of embryos at 144 hpf by a light–dark locomotion test after treatments with cabotegravir (CAB) by the immersion method. Embryos at 4 hpf were exposed until 120 hpf to CAB dissolved in fish water containing 0.1% dimethyl sulfoxide (DMSO) or fish water with 0.1% DMSO as control [CO], by the immersion method. (**A**) Movements were reported as mean ± SD total distance swam by the larvae (cm), calculated during both light and dark stimuli. (**B**) Speed (mm/s) was calculated during the same period. Results are expressed as mean ± SD of three independent experiments, with 24 larvae for each experiment and each treatment. (* *p* < 0.05 vs. control group; ** *p* < 0.005 vs. control group).

**Figure 6 ijms-24-01994-f006:**
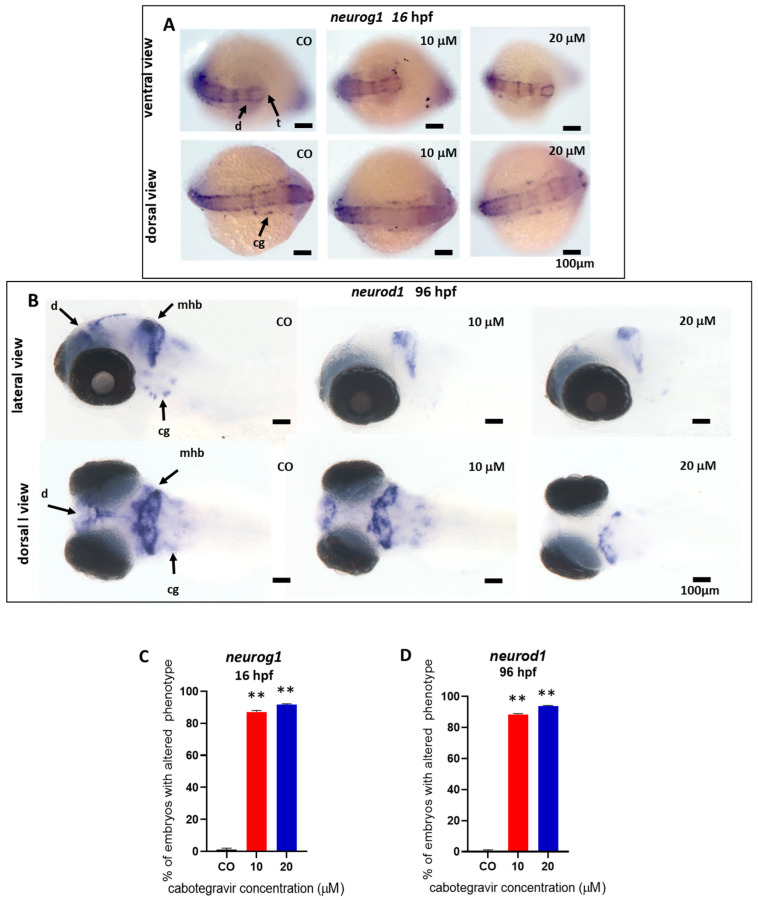
Analysis of the expression of *neurog1* and *neurod1* neural markers by whole-mount in situ hybridization (WISH) after treatment of zebrafish embryos with cabotegravir (CAB) by the immersion method. Representative images of WISH analyses of *neurog1* and *neurod1* expression in embryos treated with CAB by the immersion method. CAB was dissolved in fish water containing 0.1% dimethyl sulfoxide (DMSO). Fish water with 0.1% DMSO was used as negative control [CO]. (**A**) Ventral and dorsal views are shown for *neurog1* expression at 16 h post fertilization (hpf); (**B**) lateral and dorsal views are shown for *neurod1* expression at 96 hpf. (**C**,**D**) Percentages of embryos with defects. Experiments were performed twice with 25 embryos for each experimental point. (** *p* < 0.005 vs. control group). d, diencephalon; mhb, midbrain hindbrain boundary; cg, cranio facial ganglia; t, telencephalon.

## Data Availability

Data are available upon reasonable request to the corresponding author.

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
