# Peer review of "Cabotegravir Exposure of Zebrafish (Danio rerio) Embryos Impacts on Neurodevelopment and Behavior"

_ijms, 2023, doi:10.3390/ijms24031994_

Round 1
Reviewer 1 Report
Minor comments
Line 59-63 identifies embryo-fetal toxicities studies but studies in mice have not been mentioned. As well, there has been a recent publication on of the pregnancy outcome in humans. Please comment about mouse pregnancy models and human pregnancy outcome that has been published in support of other studies conducted (Two References are below).
Reference: Mohan H, Atkinson K, Watson B, Brumme CJ, Serghides L. A Pharmacokinetic Dose-Optimization Study of Cabotegravir and Bictegravir in a Mouse Pregnancy Model. Pharmaceutics. 2022 Aug 24;14(9):1761. doi: 10.3390/pharmaceutics14091761. PMID: 36145509; PMCID: PMC9501129.
Reference: Patel P, Ford SL, Baker M, Meyer C, Garside L, D'Amico R, Van Solingen-Ristea R, Crauwels H, Polli JW, Seal C, Yagüe Muñoz I, Thiagarajah S, Birmingham E, Spreen WR, Baugh B, van Wyk J, Vannappagari V. Pregnancy outcomes and pharmacokinetics in pregnant women living with HIV exposed to long-acting cabotegravir and rilpivirine in clinical trials. HIV Med. 2022 Nov 21. doi: 10.1111/hiv.13439. Epub ahead of print. PMID: 36411596.
As mentioned by the authors that the zebra fish is a model that could provide rapid results to understand development and toxicological, the rational of using an integrase inhibitor, that is., CAB is not clearly mentioned. Please provide clear reasoning of using this model of how it is beneficial in understanding the effects of antiretrovirals, and CAB in particular.
Please provide details of N for each methodology used in the material and methods section.
For all figures, the cabotegravir concentration below the graphs can be phrased as “Cabotegravir Concentration (µM), and the numbers in the X-axis can be shown as only 10, 20, 50, 100, 150
Figure 2B is small and the Y-axis can be split in two, to display the percentage. Please modify the image for clear depiction
Figure 3B, weight should be mentioned in (gms), not (gr). Please make changes for clarity.
Figure 5B, the thickness of the data points and error bars is not similar or consistent to the other figures presented. Please check
In the discussion, please mention clearly the dosage amount for low doses and high doses when discussing the results of significance.
Line 347-348, the phrase, “undeveloped or uninflated” font is not the same.
In the discussion, paragraph 377-391 present to be over speculative, as no studies were performed on CAB in respect to binding to Zn++. Please rephrase.

Author Response
We wish to thank the Reviewers for their useful comments and for the appreciation of our work.
We modified our manuscript accordingly. In the WORD version of the revised manuscript, we used the “Track Changes” function as suggested, such that changes can be easily viewed. A PDF clean version is also submitted for a clearer comprehension of revisions.
Below our reply point by point (reviewers’ comments in bold).
REVIEWER 1
1- Line 59-63 identifies embryo-fetal toxicities studies but studies in mice have not been mentioned. As well, there has been a recent publication on of the pregnancy outcome in humans. Please comment about mouse pregnancy models and human pregnancy outcome that has been published in support of other studies conducted (Two References are below).
Reference: Mohan H, Atkinson K, Watson B, Brumme CJ, Serghides L. A Pharmacokinetic Dose-Optimization Study of Cabotegravir and Bictegravir in a Mouse Pregnancy Model. Pharmaceutics. 2022 Aug 24;14(9):1761. doi: 10.3390/pharmaceutics14091761. PMID: 36145509; PMCID: PMC9501129.
Reference: Patel P, Ford SL, Baker M, Meyer C, Garside L, D'Amico R, Van Solingen-Ristea R, Crauwels H, Polli JW, Seal C, Yagüe Muñoz I, Thiagarajah S, Birmingham E, Spreen WR, Baugh B, van Wyk J, Vannappagari V. Pregnancy outcomes and pharmacokinetics in pregnant women living with HIV exposed to long-acting cabotegravir and rilpivirine in clinical trials. HIV Med. 2022 Nov 21. doi: 10.1111/hiv.13439. Epub ahead of print. PMID: 36411596.
Many thanks for the suggestions. We have now added comments on the suggested publications in the Discussion section adding the two suggested in the list of references (22 and 23).
2- As mentioned by the authors that the zebra fish is a model that could provide rapid results to understand development and toxicological, the rational of using an integrase inhibitor, that is., CAB is not clearly mentioned. Please provide clear reasoning of using this model of how it is beneficial in understanding the effects of antiretrovirals, and CAB in particular.
We have added a sentence near the end of the Introduction section, highlighting the importance of exploring the developmental possible effects of antiretrovirals and CAB in particular, as kindly suggested. To further emphasize the concept, we added a further sentence in the Discussion section, citing a recent review (reference 25) on the topic.
3- Please provide details of N for each methodology used in the material and methods section.
Many thanks for the suggestion. We have added the number of experiments and the number of embryos used in each experiment in the Material and Methods section, as kindly suggested. The same information is also reported in each figure legend.
4- For all figures, the cabotegravir concentration below the graphs can be phrased as “Cabotegravir Concentration (µM), and the numbers in the X-axis can be shown as only 10, 20, 50, 100, 150
Agree, many thanks. We have now modified as suggested.
5- Figure 2B is small and the Y-axis can be split in two, to display the percentage. Please modify the image for clear depiction
We have now modified as suggested. Similarly, we have modified the graph B in Supplementary Figure 3.
6- Figure 3B, weight should be mentioned in (gms), not (gr). Please make changes for clarity.
We have now modified using g as notation for grams accordingly with the MDPI Instruction for Authors (SI units).
7- Figure 5B, the thickness of the data points and error bars is not similar or consistent to the other figures presented. Please check
We have now modified as suggested.
8- In the discussion, please mention clearly the dosage amount for low doses and high doses when discussing the results of significance.
We have now modified as suggested.
9- Line 347-348, the phrase, “undeveloped or uninflated” font is not the same.
We have now modified as suggested.
10- In the discussion, paragraph 377-391 present to be over speculative, as no studies were performed on CAB in respect to binding to Zn++. Please rephrase.
We agree with the Reviewer, the cited reference regards mainly DTG, but the in-silico study with other INSTIs like CAB could suggest a similar inhibition of MMPs activity. We have modified these sentences, considering this point only as a speculation.

Reviewer 2 Report
Dear Editor,
After carefully reading the manuscript entitled Cabotegravir exposure of zebrafish (Danio rerio) embryos impacts on neurodevelopment and behavior I find it suitable for publication after only minor changes.
The introduction section clearly describes the downfall of the utilized drug for HIV infection and the necessity for further safety experiments. The materials and methods section is well organized describing all the methods performed and with citations confirming the validity of the methods. The results section is organized so that it follows the materials and methods. The discussion is concise and is not repeating the results uneceserly. The manuscript also clearly highlights the limitations of the study.
I would only advise the authors the following:
1. Please provide slightly better image quality.
2. Figure 2, graph B should be larger.
3. The Figure legends are too long, please remove the redundant information, leaving only a minimum for the reader to comperhanse the graphs.
Author Response
We wish to thank the Reviewers for their useful comments and for the appreciation of our work.
We modified our manuscript accordingly. In the WORD version of the revised manuscript, we used the “Track Changes” function as suggested, such that changes can be easily viewed. A PDF clean version is also submitted for a clearer comprehension of revisions.
Below our reply point by point (reviewers’ comments in bold).
- Please provide slightly better image quality.
We have now modified figures with (hopefully) better images. Perhaps the conversion into files compatible with the manuscript template has changed image resolution.
If needed we can send the original images to the Editorial Office.
- Figure 2, graph B should be larger.
Many thanks for the suggestion, we apologize. Accordingly, we have enlarged the graph B in Figure 2, and similarly the graph B in Supplemental Figure 3.
- The Figure legends are too long, please remove the redundant information, leaving only a minimum for the reader to comperhanse the graphs.
We kindly agree with the Reviewer. We have shortened all legends, including those of Supplemental Figures.
